# Blend-Aware Latent Diffusion: Mitigating Stitched Seams in Image Inpainting

## Abstract

Image inpainting aims to fill missing or masked regions of an image in a manner that blends with the surrounding context. While diffusion models have significantly improved the visual fidelity of inpainting, they still suffer from noticeable stitched seams, including **boundary discontinuity** and **content inconsistency** between the preserved and generated regions. We argue that these issues originate from a fundamental limitation: the latent blending of the two regions in inference, which is unaccounted for in training, creates a piece-wise latent manifold. Firstly, the masked input encoded by the Variational AutoEncoder (VAE) does not perfectly align with the resized mask, resulting in boundary discontinuity that persists in the reconstruction and denoising processes. Second, the piece-wise latent manifold deviates from the assumption of data coherence in diffusion models since the two regions follow distinct distributions, leading to content inconsistency. In this work, we propose **Blend-Aware Latent Diffusion**, a unified framework that explicitly resolves these issues by aligning the model's training dynamics with the blend nature of inference. Our framework consists of two complementary components: **BlendRecon**, a blend-aware VAE that learns to decode blended latents continuously; and **BlendGen**, a novel denoising loss that explicitly regularizes the generated content to harmonize with the surrounding context. Extensive experiments on BrushBench and MISATO demonstrate that Blend-Aware Latent Diffusion effectively mitigates stitched seams and improves perceptual quality across various scenarios, including inpainting and outpainting.

## 1 Introduction

Image inpainting Xu et al. (2023), which involves filling in missing or masked regions of an image, has long been a challenging problem in computer vision. Recent advancements in deep learning have led to the development of generative models capable of producing high-quality inpainted images Liu et al. (2020); Ntavelis et al. (2020), where diffusion models Ho et al. (2020); Song et al. (2021); Rombach et al. (2022) have gained prominence. Despite the effectiveness of diffusion-based inpainting methods Avrahami et al. (2023); Manukyan et al. (2023); Ju et al. (2024), a critical challenge remains largely

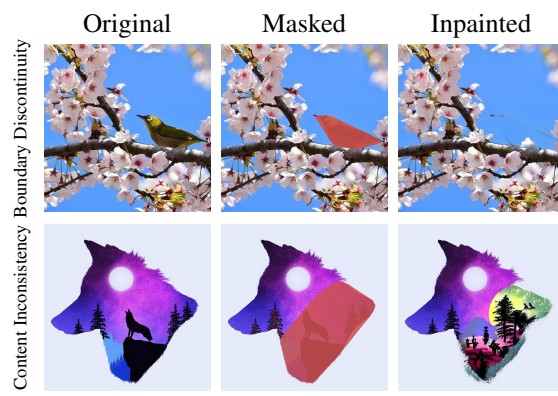

Figure 1: Stitched seams in diffusion-based inpainting.

overlooked, yet is crucial in real-world applications, that is, the stitched seams between preserved and generated regions, as shown in fig. 1.

The core of this problem lies in a structural limitation in existing pipelines. At each denoising step in inference, diffusion-based inpainting typically involves blending the generated content with the unmasked regions by copy-pasting in latent space. While this strategy effectively ensure the quality of unmasked regions, it introduces two critical challenges: (1) the resized mask introduces inaccuracies in the blending with the masked input encoded by VAE, leading to boundary discontinuity; (2) the

preserved and generated regions follow distinct statistical distributions, forming a piece-wise latent manifold with sharp transition across regions. These artifacts are not incidental defects but manifestations of a fundamental misalignment between the blend-style inference and the standard diffusion training. BrushNet Ju et al. (2024) presents a pixel-space solution for the boundary discontinuity based on a blurred mask. However, this approach may compromise the accuracy of detail preservation. BrushNet also designs a dual-branch network, where the features from the branch networks are injected into the frozen pre-trained main network layer by layer to increase the content consistency. ASUKA Wang et al. (2025) proposes a post-processing method that incorporates alignment through an external masked autoencoder and a conditional decoder to enhance content consistency.

Instead of treating stitched seams as superficial or post-generation errors to be corrected, we argue that a principled solution requires explicitly reconciling the model's learning process with the conditions of inference. This approach is both simple and effective, requiring no modifications to the network architecture or the incorporation of external models. Specifically, we propose to align the training dynamics of both the VAE and the denoiser with the piece-wise latent manifold that naturally arises in inpainting tasks.

To this end, we introduce **Blend-Aware Latent Diffusion**, a unified inpainting framework that mitigates the stitched seams through two complementary components: **BlendRecon**: a blend-aware VAE that learns to decode blended latents continuously, and **BlendGen**: a novel denoising loss that explicitly regularizes the generated content to harmonize with the surrounding context. Together, these components provide a principled, model-intrinsic solution. Our main contributions are as follows:

- We formally analyze an inherent limitation in diffusion-based inpainting: the latent blending introduces a piece-wise latent manifold, resulting in stitched seams including boundary discontinuity and content inconsistency.

- We propose Blend-Aware Latent Diffusion, a unified framework that explicitly addresses stitched seams by aligning both reconstruction and denoising processes with the blend conditions in inference.

- Extensive experiments on BrushBench and MISATO demonstrate that our method substantially mitigates stitched seams and outperforms existing inpainting models in both perceptual quality and seam visibility.

## 2 RELATED WORK

**Diffusion-based inpainting methods**. Blended diffusion Avrahami et al. (2022) and Blended Latent Diffusion (BLD) Avrahami et al. (2023) are representative diffusion-based inpainting methods, which enable inpainting masked regions by replacing unmasked regions with the original image at each denoising step in the pixel space and latent space, without altering the pre-trained diffusion model. SmartBrush Xie et al. (2023) improves the shape-guided inpainting by introducing multiple masks of the same object during training. HD-Painter Manukyan et al. (2023) enhances the text alignment in the painting region by introducing prompt-aware introverted attention. PowerPaint Zhuang et al. (2024) integrates multiple inpainting tasks into one model by introducing learnable task prompts and targeted fine-tuning strategies. Although these methods have shown promising results in structural coherence and diversity for content filling, they still display noticeable stitched seams when blending preserved regions with generated content.

**Stitched seams**. Stitched seams in diffusion-based inpainting typically manifest in three forms: color shifts, boundary discontinuity and content inconsistency. **Color shifts** occur when the original image is blended with generated content in the pixel space, which exposes the inherent reconstruction errors of the VAE. Recent works have explored various strategies to address color shifts. BLD Avrahami et al. (2023) fine-tunes the decoder to reduce pixel-level color shifts. ASUKA Wang et al. (2025) introduces a larger decoder to allow more detailed recovery. Asymmetric VQGAN Zhu et al. (2023) also redesigns the VAE decoder with an asymmetric architecture to better reconstruct details, while DiffHarmony++ Zhou et al. (2024) introduces conditional zero-convolution layers for color harmonization. **Boundary discontinuity** appears when latent blending is applied between the encoded full inputs and masked inputs in the latent denoising process. Although using the encoded full inputs alone could reduce seams, this approach risks information leakage and undermines tasks

like object inapinting and object removal. BrushNet Ju et al. (2024) attempts to alleviate this issue by applying a blurred mask in pixel space. However, this leads to a loss of structural detail near the boundary. **Content inconsistency** refers to the failure of the generated region to align structurally and semantically with the surrounding context. BrushNet designs a dual-branch network that separates the masked image features and noisy latent vectors into different branches. The features from the branch networks are injected into the frozen pre-trained main network layer by layer, which increases the content consistency. ASUKA introduces alignment based on an external masked autoencoder and a conditional decoder to improve content consistency. In contrast, boundary discontinuity and content inconsistency have received limited attention in prior works, and our method is designed to explicitly address both.

## 3 METHOD

In this section, we present **Blend-Aware Latent Diffusion**, a unified framework designed to explicitly address stitched seams in diffusion-based inpainting.

### 3.1 PRELIMINARIES

Denoising diffusion probabilistic models Ho et al. (2020) aim to transform pure noise $\mathbf{x}_T$ into a coherent output image $\hat{\mathbf{x}}_0$, guided by the given conditioning prompt. In the forward process, noise is added to a clean image:

$$\mathbf{x}_t = \sqrt{\bar{\alpha}_t}\mathbf{x}_0 + \sqrt{1 - \bar{\alpha}_t}\epsilon \tag{1}$$

where $\mathbf{x}_t$ denotes the noisy image at timestep $t$, $\bar{\alpha}_t$ are hyper-parameters governing the noise schedule over $t \in [1, T]$. During training, the network $\theta$ is optimized to predict the noise $\epsilon$ given the noisy image $\mathbf{x}_t$:

$$\min_\theta \mathbb{E}_{\mathbf{x}_0,\epsilon \sim \mathcal{N}(0,I), t \sim \mathcal{U}(1,T)} \|\epsilon - \epsilon_\theta(\mathbf{x}_t, t, \mathcal{C})\|_2^2, \tag{2}$$

where $\mathcal{C}$ is the conditioning input. Stable diffusion Rombach et al. (2022) further incorporates a Variational Auto-Encoder (VAE) to map the input image into a lower-dimensional latent space and significantly reduce computational cost. In this setting, the latent representation $\mathbf{z}_0$ is obtained as $E(\mathbf{x}_0)$, where $\mathbf{x}_0$ is the input image, and the reconstructed output image is derived as $\hat{\mathbf{x}}_0 = D(\hat{\mathbf{z}}_0)$. To enhance performance in low-SNR steps, v-prediction Salimans & Ho (2022) is employed and the targets are formulated as:

$$\mathbf{v} = \sqrt{\bar{\alpha}_t}\epsilon - \sqrt{1 - \bar{\alpha}_t}\mathbf{z}_0. \tag{3}$$

The original latent $\mathbf{z}_0$ can then be approximated by:

$$\hat{\mathbf{z}}_0 = \sqrt{\bar{\alpha}_t}\mathbf{z}_t - \sqrt{1 - \bar{\alpha}_t}\mathbf{v}_\theta(\mathbf{z}_t) \tag{4}$$

where $\mathbf{z}_t = \sqrt{\bar{\alpha}_t}\mathbf{z}_0 + \sqrt{1 - \bar{\alpha}_t}\epsilon$. Our work builds upon the stable diffusion model architecture.

### 3.2 PROBLEM ANALYSIS

Blended Latent Diffusion (BLD) is a widely used method and often serves as the default inpainting approach. Given an original image $\mathbf{x}_0$ and a binary mask $\mathbf{M}$, where 0 represents the regions to be inpainted. The masked image is $\mathbf{x}_0^{\mathbf{M}} = \mathbf{x}_0 \odot \mathbf{M}$. BLD begins by extracting the latent using VAE: $\mathbf{z}_0^{\mathbf{M}} = E(\mathbf{x}_0^{\mathbf{M}})$. Subsequently, the mask M is resized to $\mathbf{m}$ to match the size of $\mathbf{z}_0^{\mathbf{M}}$. In denoising, BLD applies noise to $\mathbf{z}_0^{\mathbf{M}}$ up to the desired noise level at timestep $t$, yielding $\mathbf{z}_t^{\mathbf{M}}$. At the following denoising steps, the latent is blended:

$$\mathbf{z}_{t-1}^{\text{blend}} = (1 - \mathbf{m}) \odot \mathbf{z}_{t-1}^{\text{denoise}} + \mathbf{m} \odot \mathbf{z}_{t-1}^{\mathbf{M}} \tag{5}$$

$\mathbf{z}_{t-1}^{\text{denoise}}$ means the latent is obtained from the denoising process. The blending operation preserves the unmasked content while allowing for new content generation in the masked regions. However, it fails to guarantee coherence, as the resized mask $\mathbf{m}$ is not precisely match the invisible latent in $\mathbf{z}_t^{\mathbf{M}}$, resulting in the boundary discontinuity, i.e., $E(\mathbf{x}_0 \odot \mathbf{M}) \neq E(\mathbf{x}_0) \odot \mathbf{m}$, especially near the boundary of $\mathbf{m}$. Consequently, the discontinuity persists throughout the denoising process and the pretrained VAE ultimately reconstructs from the stitched latent. As illustrated in fig. 2, the reconstructed image shows visible seams, indicating a mismatch between the blended latents

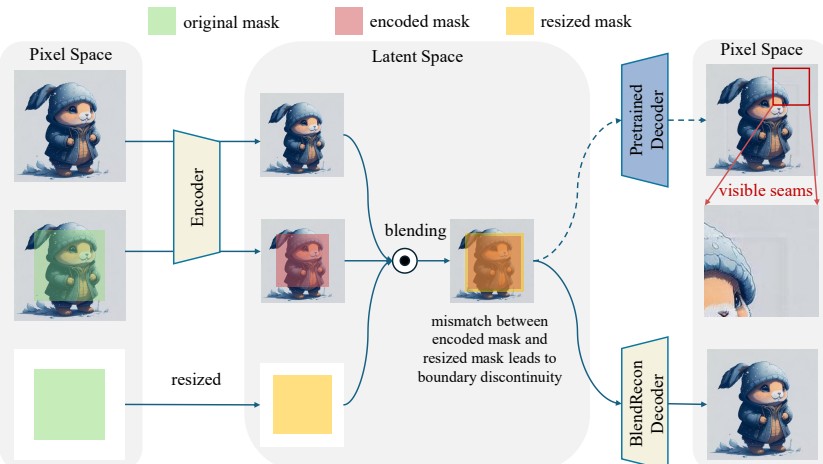

Figure 2: The reconstruction under blending. The pretrained VAE reconstructs the boundary discontinuity, whereas BlendRecon achieves a much smoother reconstruction.

and the original reconstruction objective. Additionally, the blending also impacts content consistency in generation, since the preserved and generated regions follow distinct distributions, that is, $\mathbf{z}^{\text{preserved}} \sim p_{\text{data}}(\mathbf{z})$, $\mathbf{z}^{\text{generated}} \sim p_{\text{denoise}}(\mathbf{z})$. Therefore, we argue that the latent blending creates a piece-wise latent manifold and introduces artifacts in both the reconstruction and denoising, manifesting as boundary discontinuity and content inconsistency near the mask boundary.

**Theoretical insight**. We formalize the issues through the following propositions.

**Proposition 1 (Boundary Discontinuity).** Assume the VAE decoder $D$ is Lipschitz continuous with constant $L$, i.e., $\|D(\mathbf{z}_1) - D(\mathbf{z}_2)\| \leq L \cdot \|\mathbf{z}_1 - \mathbf{z}_2\|$. Then the reconstruction discrepancy near the mask boundary is bounded by $\|D(\mathbf{z}^{\text{blend}}) - D(\mathbf{z})\|_\Omega \leq L \cdot \|\mathbf{z}^{\text{blend}} - \mathbf{z}\|_{\Omega,}$, where $\|\cdot\|_\Omega$ denotes the masked $L_2$ norm restricted to pixels in the boundary band $\Omega$. Since the VAE reconstruction of the original image is not exact, we further have $\|D(\mathbf{z}^{\text{blend}}) - \mathbf{x}\|_\Omega \leq L \cdot \|\mathbf{z}^{\text{blend}} - \mathbf{z}\|_\Omega + \epsilon_{\text{VAE},}$, with $\epsilon_{\text{VAE}} = \|D(\mathbf{z}) - \mathbf{x}\|_\Omega$ denoting the reconstruction error of $D$. Therefore, due to the mismatch $E(\mathbf{x} \odot \mathbf{M}) \neq E(\mathbf{x}) \odot \mathbf{m}$, the boundary discontinuity will be reconstructed by the VAE, leading to visible seams. As latent blending is inevitable, this motivates us to explicitly train the model to recognize and correct such artifacts.

**Proposition 2 (Content Inconsistency).** Let $f_\theta$ denote the denoiser parameterized by $\theta$, which predicts the velocity $\mathbf{v}$ in $\mathbf{v}$-prediction. According to eq. (5), there exists a distributional divergence between preserved and generated regions: $p_{\text{data}}(\mathbf{z}) \neq p_{\text{denoise}}(\mathbf{z})$. Without supervision, the alignment between the two regions is under-constrained and the denoiser cannot bridge this divergence, which highlights the need to explicitly blend during training. By blending the predicted result and re-predicting the denoising direction, boundary generation is calibrated and constrained: $f_\theta(\mathbf{z}_t^{\text{blend}}, t)_\Omega \approx \mathbf{x}_\Omega$. This explicitly connects the denoiser output to the true target and encourages content alignment and helps the model learn smooth transitions.

## 3.3 BLEND-AWARE LATENT DIFFUSION FRAMEWORK

To mitigate stitched seams, we propose blend-aware latent diffusion, which introduces blended training into both reconstruction and denoising.

### 3.3.1 BLENDRECON: BLEND-AWARE VAE

As discussed above and shown in fig. 2, blending in latent space can lead to boundary discontinuity, since the VAE trained to reconstruct from full latents is not inherently suited for handling blended latents. Specifically, standard reconstruction training optimizes:

$$\min_{\theta_{\text{VAE}}} \mathbb{E}_{\mathbf{x}} \left[ \|D_{\theta_{\text{VAE}}}(E(\mathbf{x})) - \mathbf{x}\|^2 \right] \tag{6}$$

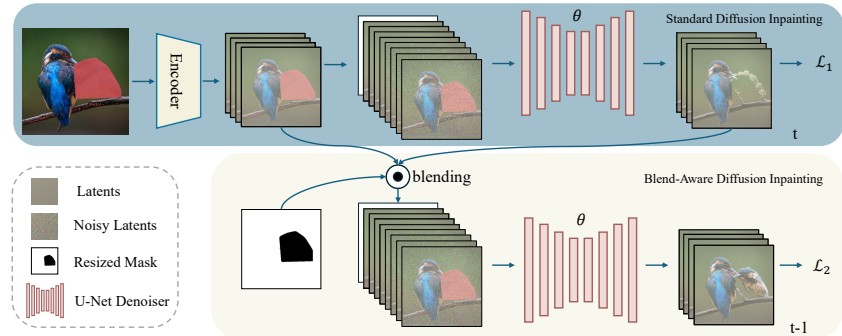

Figure 3: The illustration of BlendGen. $\mathcal{L}_1$ represents the standard diffusion loss, where the model predicts $\mathbf{z}_0$. After obtaining $\hat{\mathbf{z}}_0$, we simulate the blending operation and calculate the latent for the next timestep, making the model predict the noise to calculate $\mathcal{L}_2$. The $\theta$ in the figure represents the same denoiser during training.

This objective fails to account for latent compositions arising from copy-paste blending in inference. To address this, we fine-tune the VAE with the following objective:

$$\min_{\theta_{\text{VAE}}} \mathbb{E}_{\mathbf{x},\mathbf{m}} \left[ \left\| D_{\theta_{\text{VAE}}} \big( (1 - \mathbf{m}) \odot E(\mathbf{x}) + \mathbf{m} \odot E(\mathbf{x} \odot \mathbf{M}) \big) - \mathbf{x} \right\|^2 \right] \tag{7}$$

which enables the decoder to learn seamless reconstructions of blended latents, thereby reducing boundary discontinuity.

### 3.3.2 BLENDGEN: BLEND-AWARE DENOISER

In standard diffusion training, the denoiser learns to predict noise from noisy samples drawn from the data distribution, under the assumption of global data coherence. However, diffusion-based inpainting deviates from this assumption: at each denoising step, the latent is blended by copy-pasting preserved regions from the original image, resulting in a distributional divergence unaccounted for during training. Moreover, vanilla inpainting methods cannot ensure that the generated content will be confined exactly within the specified mask region, since the only 'constrain' is that the gradual diffusion process masking and blending encourages this behavior. Similarly, they also have no constraints to align the generated content with the preserved content, leading to content inconsistency. To overcome this, we introduce explicit constraints through a regularized loss that aligns the regions.

In the conventional v-prediction diffusion training, the objective is to predict the velocity $\mathbf{v}$ from the latent $\mathbf{z}_t$. During inference, after the model predicts the velocity, we compute and blend $\hat{\mathbf{z}}_{t-1}$ iteratively. For BlendGen in algorithm 1, besides the standard diffusion training, we further compute $\hat{\mathbf{z}}_0$ using the predicted $\mathbf{v}_\theta$, followed by a blending to update $\hat{\mathbf{z}}_0$. It's important to note that, while the blending in inference is applied to $\mathbf{z}_t$, during training, it's performed on $\hat{\mathbf{z}}_0$. Since $\hat{\mathbf{z}}_{t-1}$ is calculated by adding noise to $\hat{\mathbf{z}}_0$, blending $\hat{\mathbf{z}}_0$ is effectively equivalent to blending $\hat{\mathbf{z}}_{t-1}$. Given that $\hat{\mathbf{z}}_0$ is not a real sample, we calculate the true noise $\epsilon_2^*$ to adjust the inpainting accordingly. With $\epsilon_2^*$, we derive the target $\mathbf{v}_2$ for the blended objective. Finally, the model parameters $\theta$ are updated based on the combined losses $\mathcal{L}_1 + \lambda \mathcal{L}_2$, where $\lambda$ serves as a weight factor to balance the two loss terms.

---

**Algorithm 1:** Blend-Aware Denoiser

**Input:** denoise parameters $\theta$, noises $\epsilon_1, \epsilon_2$, sample $\mathbf{z}_0$, original mask $\mathbf{M}$, resized mask $\mathbf{m}$, timestep $t$, conditioning input $\mathcal{C}$

**Output:** optimized parameters $\theta$

1   $\mathbf{z}_t = \sqrt{\bar{\alpha}_t}\mathbf{z}_0 + \sqrt{1 - \bar{\alpha}_t}\epsilon_1$;

2   $\mathbf{v}_1 = \sqrt{\bar{\alpha}_t}\epsilon_1 - \sqrt{1 - \bar{\alpha}_t}\mathbf{z}_0$;

3   $\mathcal{L}_1 = \|\mathbf{v}_1 - \mathbf{v}_\theta(\mathbf{z}_t, t, \mathcal{C})\|_2^2$;

4   $\hat{\mathbf{z}}_0 = \sqrt{\bar{\alpha}_t}\mathbf{z}_t - \sqrt{1 - \bar{\alpha}_t}\mathbf{v}_\theta(\mathbf{z}_t, t, \mathcal{C})$;

5   Blending $\hat{\mathbf{z}}_0 = \hat{\mathbf{z}}_0 \odot (1 - \mathbf{m}) + \mathbf{z}_0^{\mathbf{M}} \odot \mathbf{m}$;

6   $\hat{\mathbf{z}}_{t-1} = \sqrt{\bar{\alpha}_{t-1}}\hat{\mathbf{z}}_0 + \sqrt{1 - \bar{\alpha}_{t-1}}\epsilon_2$;

7   Calculate the true noise $\epsilon_2^* = \frac{\hat{\mathbf{z}}_{t-1} - \sqrt{\bar{\alpha}_{t-1}}\mathbf{z}_0}{\sqrt{1 - \bar{\alpha}_{t-1}}}$;

8   $\mathbf{v}_2 = \sqrt{\bar{\alpha}_{t-1}}\epsilon_2^* - \sqrt{1 - \bar{\alpha}_{t-1}}\mathbf{z}_0$;

9   $\mathcal{L}_2 = \|\mathbf{v}_2 - \mathbf{v}_\theta(\hat{\mathbf{z}}_{t-1}, t-1, \mathcal{C})\|_2^2$;

10   $\theta \leftarrow \theta - \eta \nabla_\theta(\mathcal{L}_1 + \lambda \mathcal{L}_2)$;

11   **return** $\theta$

---

$\mathcal{L}_2$ encourages the model to explicitly learn how to handle blended latents, enabling better content consistency between the two regions throughout the diffusion process.

| Original | Masked | Blend Diffusion | HD-Painter | PowerPaint | BrushNet | Ours |
|---|---|---|---|---|---|---|

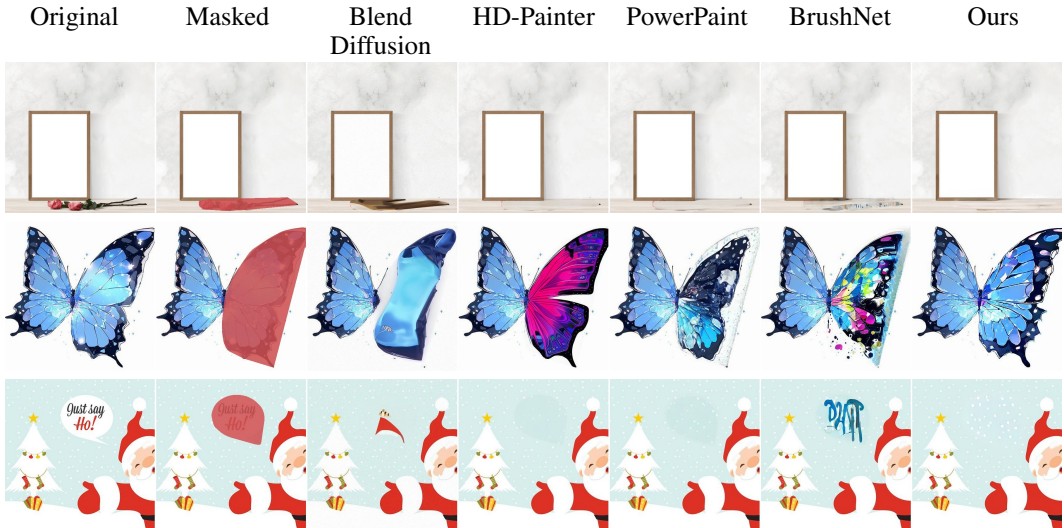

Figure 4: Comparison of boundary continuity between our method and other approaches for inpainting.(We encourage readers to zoom in to better observe the details.)

## 4 EXPERIMENTS

In this section, we evaluate the proposed method qualitatively and quantitatively, for inpainting and outpainting.

**Implementation details.** In our experiments, unless otherwise specified, all tasks are based on Stable Diffusion 1.5 (SD-1.5). The VAE is trained using the default training settings, consistent with those in the SD-1.5 VAE training phase. Apart from splitting the original VAE input into two parts, the VAE model structure remains unchanged. Specifically, we blended the two input parts in the latent space and reconstructed the blended latents. For the denoiser, we set the number of sampling steps to be 50 by using a stride of 20 over 1000 diffusion steps with a guidance scale of 3. The loss weight $\lambda$ is set to 0.5 by default. For the inpainting, we utilize the saliency detection model U2Net Qin et al. (2020) to segment out the background, then randomly sample masks within the background as input during training. For outpainting, we apply a mask around the edges of the training images, retaining only the central portion. In both tasks, the training objective of the denoiser is to reconstruct the original images.

**Dataset.** We filter the publicly available Laion-Aesthetic Schuhmann et al. (2022) dataset, selecting images with a resolution of at least 512x512 as our training set (approximately 3 million images). Both the VAE and denoiser are fine-tuned based on this dataset.

**Evaluation benchmarks.** To comprehensively evaluate performance across diverse scenarios, we employ BrushBench Ju et al. (2024) and MISATO Wang et al. (2025) with a resolution of 512x512. BrushBench contains 600 images, covering multiple categories (e.g., humans, animals, outdoor). MISATO offers 2,000 samples selected from representative datasets: Matterport3D Chang et al. (2017), MegaDepth Li & Snavely (2018), Flickr Lin et al. (2022), and COCO 2014 Lin et al. (2014), capturing varied domains (e.g., landscape, indoor, building and background) and mask styles. To reduce the strong priors in BrushBench's segmentation-based masks, we extract and slightly expand their convex hulls to generate smoother masks without preserving object shapes. For inpainting tasks guided by prompts, we use 'scenery', which is the best one shown in PowerPaint, as the default prompt.

**Evaluation metrics.** Our evaluation focuses on two aspects: image quality and seam visibility. We utilize Fréchet Inception Distance (FID) Heusel et al. (2017), Aesthetic Score (AS) Schuhmann et al. (2022) and Learned Perceptual Image Patch Similarity (LPIPS) Zhang et al. (2018) to measure

Figure 5: Comparison of content consistency between our method and other approaches for inpainting.

the quality. In addition, we propose a new metric to measure stitched seams: given an inpainted image $\hat{\mathbf{x}}$ and the original image $\mathbf{x}$, we define the Seam Visibility (SV) as the average L2 distance of RGB values between pixels along the mask boundary and their neighboring pixels. It can be defined as:

$$\text{SV} = \frac{1}{|\Omega|} \sum_{(i,j) \in \Omega} \|\hat{\mathbf{x}}_{i,j} - \mathbf{x}_{i,j}\|_2 \tag{8}$$

Where $\Omega$ denotes the pixels within a narrow band near the mask boundary, $\hat{\mathbf{x}}_{i,j}$ and $\mathbf{x}_{i,j}$ means the RGB values of inpainted images and original images at $(i,j)$ separately. In practice, we set the band width to 3 pixels on both sides of the mask boundary, which is sufficient to capture visible discontinuities while avoiding interference from regions far from the seam. We also normalize the SV by the length of the mask boundary, ensuring that the metric remains comparable across small and large masked regions. We also conducted sensitivity analysis with varying band widths (e.g., 3–7 pixels) and observed stable results, further confirming the robustness of this metric.

### 4.1 QUALITATIVE EVALUATION

We conduct extensive experiments for the qualitative comparison between our method and SOTA approaches. We keep the recommended hyperparameters for each inpainting method in all images for fair comparison.

We present the visual results of different methods on the inpainting task, from two perspectives, boundary continuity and content consistency, as shown in fig. 4 and fig. 5, respectively. In fig. 4, existing inpainting methods always exhibit boundary discontinuity (zoom in to observe the details), while our method demonstrates excellent continuity. In inpainting tasks, reasonable semantics and structure are crucial to create a cohesive, visually complete image. Our method generates images with more coherent and realistic structures than other methods, as presented in fig. 5.

We also visualize the results of different methods on the outpainting task in fig. 6. In the transition area between the non-masked and generated regions, the compared methods show noticeable stitched seams, whereas our results do not suffer from this issue. In terms of content and structure, BrushNet and HD-painter tend to produce frames around the image. Compared to PowerPaint, our method demonstrates advantages in boundary continuity and structure coherence, such as the generation of human body structures in the first column and the moon in the second column. Consequently, for both inpainting and outpainting tasks, our approach demonstrates clear advantages in both boundary continuity and content consistency.

Figure 6: Comparison of stitched seams between our method and other approaches for outpainting.

| Task | Methods | BrushBench | | | | MISATO | | | |
|------|---------|-----------|------|----------|------|--------|------|----------|------|
| | | FID($\downarrow$) | AS($\uparrow$) | LPIPS($\downarrow$) | SV($\downarrow$) | FID($\downarrow$) | AS($\uparrow$) | LPIPS($\downarrow$) | SV($\downarrow$) |
| Inpainting | BLD Avrahami et al. (2023) | 18.466 | 5.367 | 0.083 | 15.735 | 16.324 | 5.396 | 0.075 | 17.191 |
| | HD-Painter Manukyan et al. (2023) | 15.604 | 5.985 | 0.061 | 19.029 | 13.859 | 5.410 | 0.050 | 16.315 |
| | PowerPaint Zhuang et al. (2024) | 15.848 | 5.550 | 0.029 | 12.754 | 14.538 | 5.714 | 0.034 | 12.436 |
| | BrushNet Ju et al. (2024) | 17.824 | 5.859 | 0.037 | 21.036 | 14.699 | 5.663 | 0.039 | 15.482 |
| | **Ours** | **15.434** | **5.991** | **0.024** | **8.538** | **13.359** | **5.879** | **0.021** | **11.634** |
| Outpainting | HD-Painter Manukyan et al. (2023) | 30.596 | 5.897 | 0.503 | 14.871 | 25.489 | 5.691 | 0.475 | 12.589 |
| | PowerPaint Zhuang et al. (2024) | 22.768 | 5.606 | 0.419 | 12.389 | 18.721 | 5.880 | 0.241 | 11.119 |
| | BrushNet Ju et al. (2024) | 30.342 | 5.487 | 0.485 | 15.458 | 27.212 | 5.462 | 0.519 | 13.675 |
| | **Ours** | **19.530** | **6.131** | **0.361** | **10.249** | **15.757** | **5.963** | **0.192** | **10.529** |

Table 1: Quantitative evaluation of different methods on BrushBench and MISATO.

## 4.2 QUANTITATIVE EVALUATION

We have also conducted a comprehensive quantitative evaluation. The results, presented in table 1, clearly demonstrate that our approach significantly outperforms other methods across multiple evaluation metrics. Specifically, our method achieves superior results in both image quality and seam visibility, effectively reducing visible artifacts at the boundary. The outpainting results validate our method's ability to handle larger context expansions beyond the masked regions, demonstrating robustness and flexibility. Given BLD's relatively limited performance on the outpainting, we exclude it from comparisons in this setting for a fairer evaluation.

To further evaluate the effectiveness of our method in producing visually coherent and seamless results, we conducted a user study. The study involved 20 participants. We randomly sample test images with all methods

| Tasks | BLD | HD-Painter | PowerPaint | BrushNet | Ours |
|-------|-----|-----------|-----------|----------|------|
| Inpainting | 2% | 32% | 10% | 10% | 46% |
| Outpainting | - | 4% | 14% | 0% | 82% |

Table 2: A user study to compare different methods.

and ask them to choose the most satisfying results per target. Everyone evaluates 100 images, 50 images for inpainting and 50 images for outpainting. The results are shown in table 2. It shows that our results are consistently preferred in both tasks.

## 4.3 ABLATION STUDY

In our method, both the BlendRecon and the BlendGen are designed to enhance image quality. The BlendRecon primarily addresses boundary discontinuity, ensuring that the generated region blends smoothly with the original image. On the other hand, the BlendGen ensures content consistency, maintaining structural coherence between the generated and original contents. To validate the effectiveness of these components, we conduct a series of ablation studies, demonstrating the individual contributions to the overall performance.

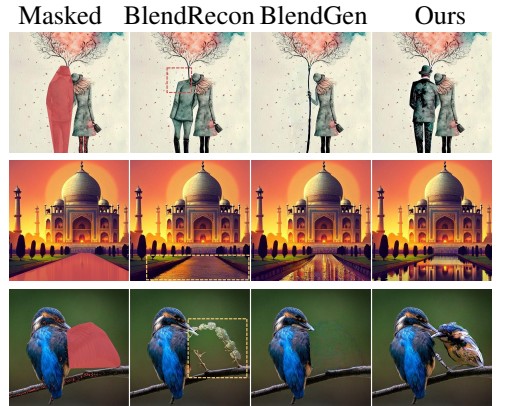

Figure 7: Ablation study for inpainting.     Figure 8: Ablation study for outpainting.

| Tasks | Methods | FID($\downarrow$) | AS($\uparrow$) | LPIPS($\downarrow$) | SV($\downarrow$) |
|---|---|---|---|---|---|
| Inpainting | Base | 18.703 | 5.537 | 0.046 | 17.093 |
| | Base+BlendRecon | 16.835 | 5.725 | 0.031 | 15.354 |
| | Base+BlendGen | 17.081 | 5.576 | 0.026 | 16.080 |
| | **Ours** | **15.434** | **5.991** | **0.024** | **8.538** |
| Outpainting | Base | 22.681 | 5.549 | 0.533 | 14.351 |
| | Base+BlendRecon | 21.523 | 5.738 | 0.451 | 12.130 |
| | Base+BlendGen | 20.649 | 5.894 | 0.411 | 11.537 |
| | **Ours** | **19.530** | **6.131** | **0.361** | **10.249** |

Table 3: Ablation study of our methods on BrushBench.

As shown in fig. 7, if only the BlendGen is used, noticeable boundary discontinuity arises, especially in cases where the mask covers a solid-color background (as seen in rows 1 and 3). On the other hand, if only the BlendRecon is applied, the generated content achieves boundary continuity but often appears sparse or lacks coherent integration with the original image in terms of structure and detail. Our proposed method, which combines both BlendRecon and BlendGen, produces images of better quality. This combined approach ensures seamless blending, where the inpainted regions not only match the boundary of the original image but also align with its structural and semantic requirements. In particular, our method generates reasonable content, including the human body, reflection, and a bird. table 3 shows the quantitative results of the two components. It can be observed that each component positively contributes to the final results. We also conduct ablation studies on the outpainting, as shown in fig. 8. The conclusions are the same as those observed in inpainting: BlendRecon primarily improves boundary continuity, while the BlendGen enhances the consistency between the preserved and generated content. It can be seen that our method avoids generating abnormal objects and correctly extends tree trunks.

## 5 CONCLUSION

In this work, we dive into the stitched seams in diffusion-based image inpainting/outpainting and analyze the underlying causes. We argue that latent blending creates a mismatched mask gap and a piece-wise manifold, resulting in boundary discontinuity and content inconsistency. To address the above issues, we propose two key solutions: BlendRecon ensures boundary continuity by enabling the VAE to correct mismatched mask gaps, while the BlendGen simulates the blending operation during training, leading to smoother generation for image content. Extensive experiments demonstrate the effectiveness and robustness of our approaches, indicating the potential to be widely applied in tasks requiring seamless integration.

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
