# BLEND-AWARE LATENT DIFFUSION: MITIGATING STITCHED SEAMS IN IMAGE INPAINTING

## A   MORE QUALITATIVE RESULTS

We present additional results based on SD-1.5, as shown in fig. 9, fig. 10, fig. 11 and fig. 12. These figures illustrate the comparative results of inpainting and outpainting tasks against other methods.

054
055
056
057
058
059
060
061
062
063
064
065
066
067
068
069
070
071
072
073
074
075
076
077
078
079
080
081
082
083
084
085
086
087
088
089
090
091
092
093
094
095
096
097
098
099
100
101
102
103
104
105
106
107

| Original | Masked | Blend Diffusion | HD-Painter | PowerPaint | BrushNet | Ours |
|---|---|---|---|---|---|---|

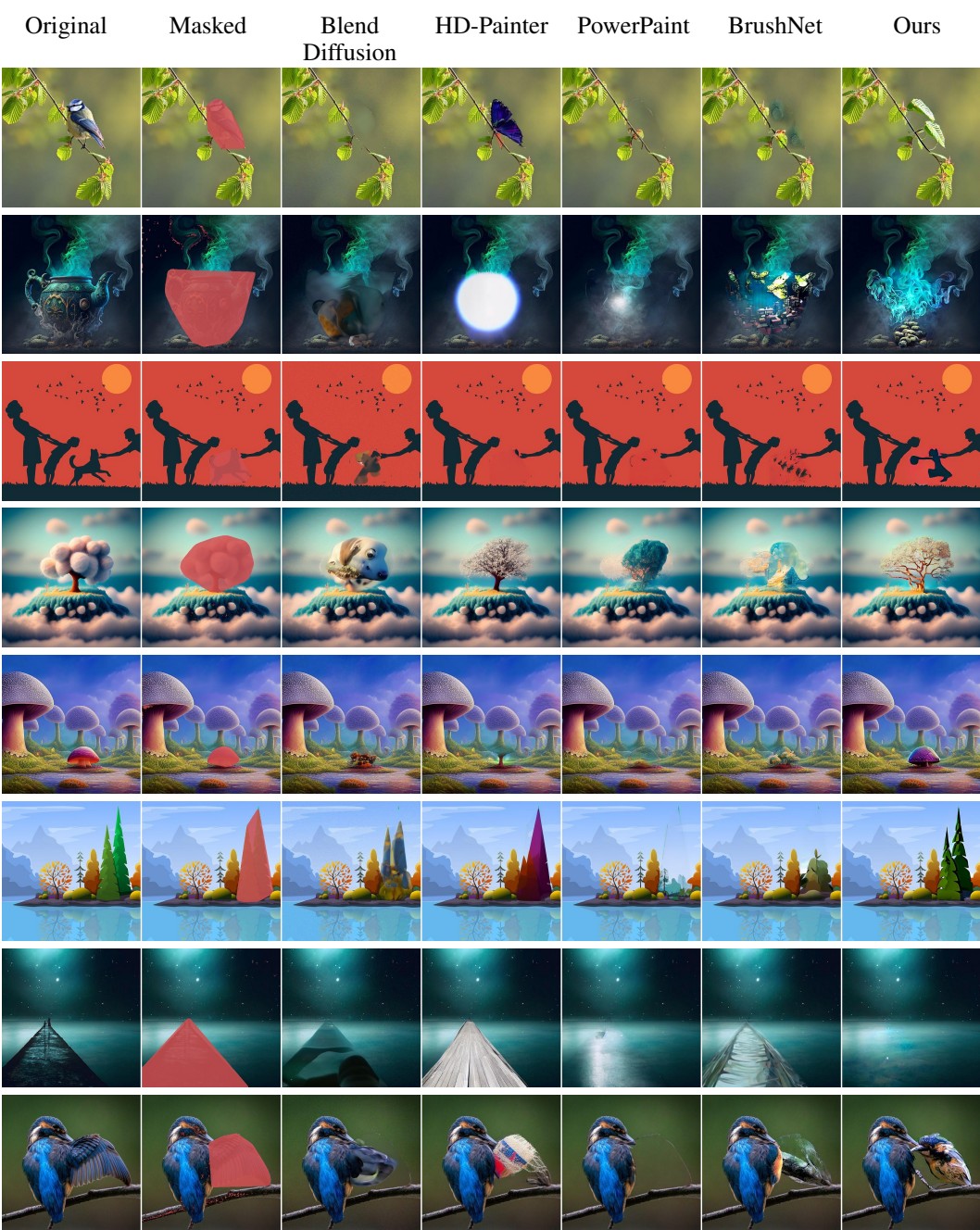

Figure 9: More comparison of boundary continuity between our method and other approaches for inpainting.(We encourage readers to zoom in to better observe the details.)

Original Masked Blend Diffusion HD-Painter PowerPaint BrushNet Ours

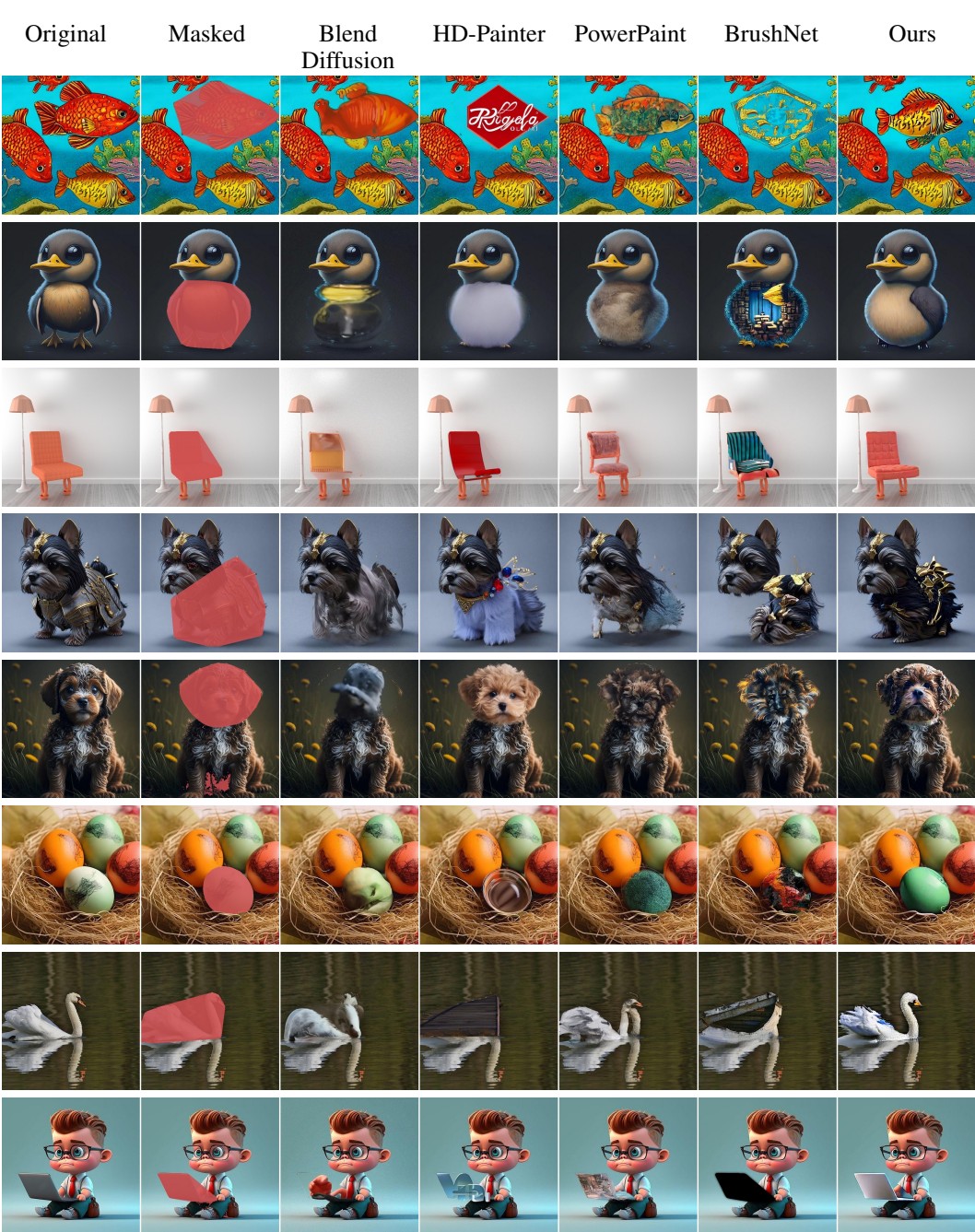

Figure 10: More comparison of content consistency between our method and other approaches for inpainting.

Masked HD-Painter PowerPaint BrushNet Ours

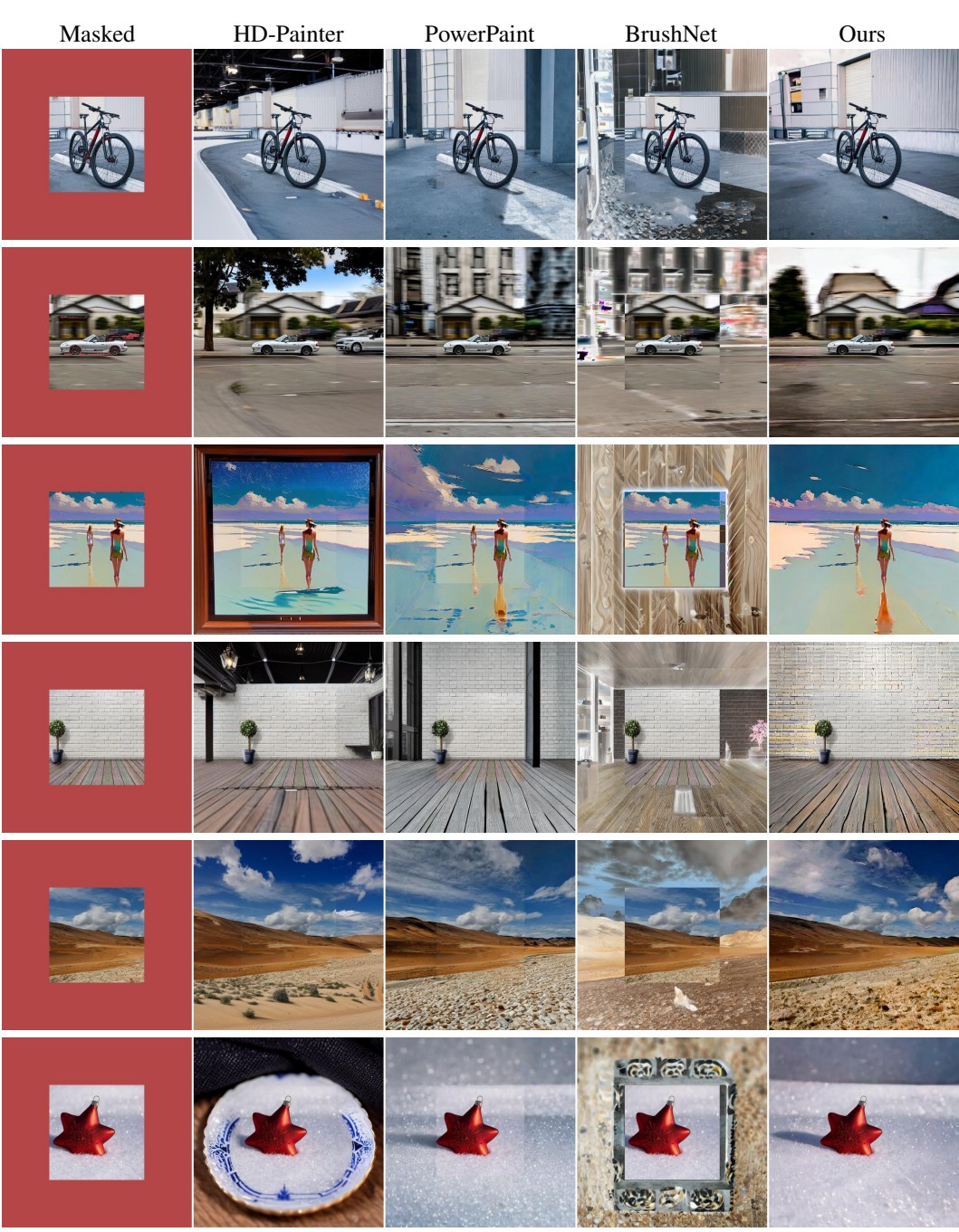

Figure 11: More comparison of boundary continuity between our method and other approaches for outpainting.(We encourage readers to zoom in to better observe the details.)

| Masked | HD-Painter | PowerPaint | BrushNet | Ours |
| --- | --- | --- | --- | --- |

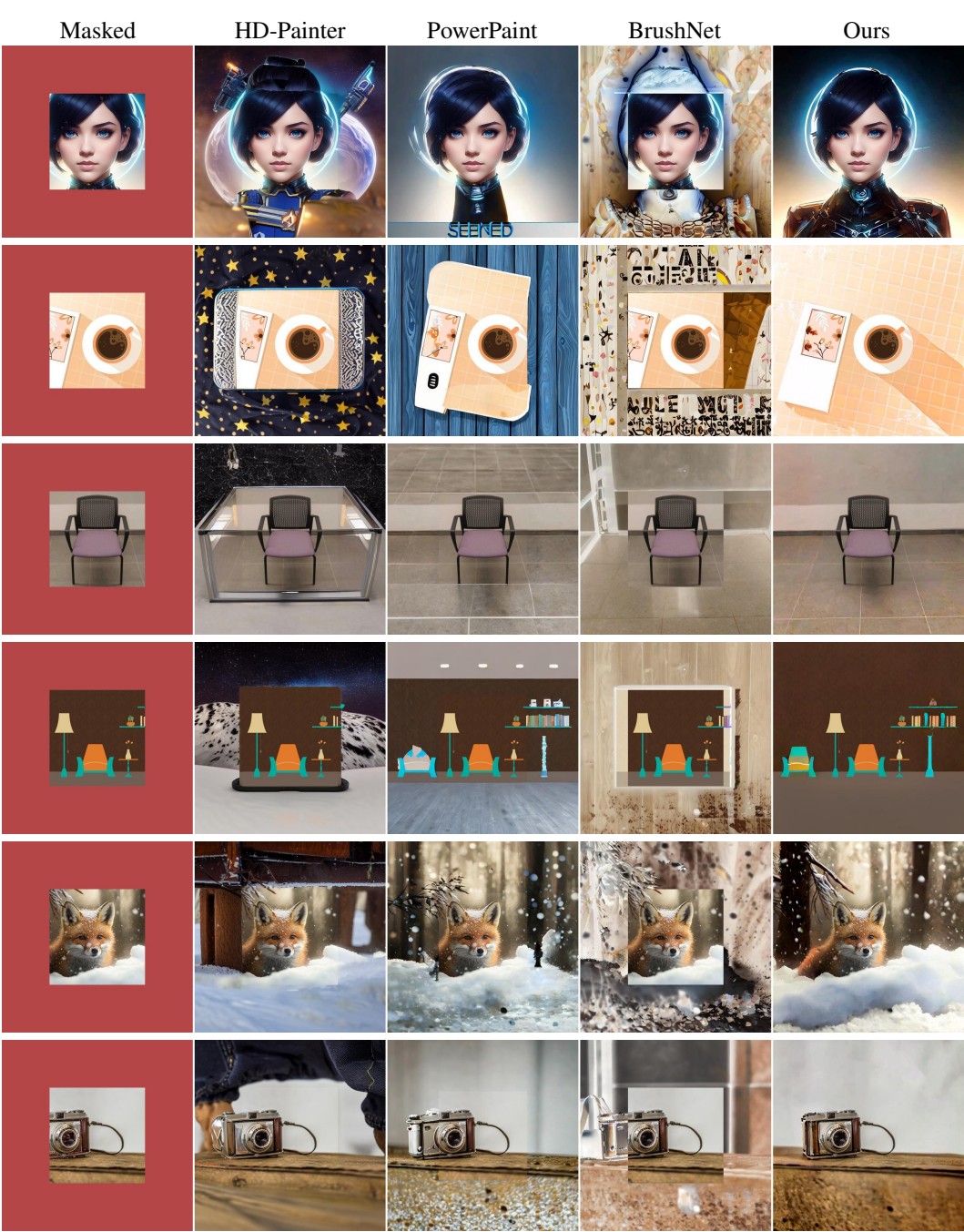

Figure 12: More comparison of content consistency between our method and other approaches for outpainting.