# OpenReview forum: "Blend-Aware Latent Diffusion: Mitigating Stitched Seams in Image Inpainting"
_ICLR.cc/2026/Conference — ICLR 2026 Conference Withdrawn Submission_

### Official Review · Reviewer_cbiB · 2025-10-29

**Soundness:** 1
**Presentation:** 2
**Contribution:** 1
**Rating:** 2
**Confidence:** 5

**Summary:**

In this work, the authors dive into the stitched seams in diffusion-based image inpainting/outpainting and analyze the underlying causes. We argue that latent blending creates a mismatched mask gap and a piece-wise manifold, resulting in boundary discontinuity and content inconsistency. To address the above issues, the authors propose two key solutions: BlendRecon ensures boundary continuity by enabling the VAE to correct mismatched mask gaps, while the BlendGen simulates the blending operation during training, leading to smoother generation for image content. Extensive experiments demonstrate the effectiveness and robustness of our approaches, indicating the potential to be widely applied in tasks requiring seamless integration.

**Strengths:**

Easy to read and a promising extension of image inpainting.

**Weaknesses:**

- The motivation of this paper is rather unclear. I believe that the two challenges proposed by the authors actually stem from improper dataset processing:

  - Regarding the first challenge in the first row of Fig. 1, after downloading and examining the BrushBench dataset, I found that the issue (Boundary Discontinuity) actually arises from the dataset itself. Specifically, during the creation of BrushBench, only the main objects were segmented, while the edge details were not carefully processed, leading to boundary discontinuities. If a more appropriate dataset, such as EditBench, were used, this problem might not even exist. Therefore, I strongly question the validity of this “challenge.” Moreover, the authors did not empirically verify that their proposed method can resolve this issue—the experimental results mainly show regular mask occlusions or occlusions with relatively simple surrounding details. This suggests that the so-called challenge is not inherently related to image inpainting itself, but rather caused by the improper dataset construction.

  - For the second challenge in the second row of Fig. 1, namely Content Inconsistency, I agree that this problem can exist. However, I believe the authors have conflated the challenges of traditional image inpainting with those of text-guided image inpainting. This challenge seems more reasonable to propose and address within the context of traditional inpainting. In text-guided inpainting, the masked regions typically remove an entire object along with its boundary areas, whereas traditional inpainting emphasizes arbitrary occlusions across the whole image, where maintaining local consistency with surrounding content is more critical. Therefore, the example shown in Fig. 1 appears to be just an isolated case from BrushBench. Consequently, the comparison with text-guided inpainting methods is highly questionable. Furthermore, the experiments fail to convincingly demonstrate that the proposed approach effectively resolves the Content Inconsistency problem—for instance, in the third row of Fig. 7, I do not consider the emergence of a new bird in the inpainted region to be an optimal or reasonable outcome.

- Blend-Aware Denoiser process lacks necessary visualizations, making it unclear how the L2 term addresses the issue of content inconsistency during the denoising process. The experimental section fails to provide an analysis of computational complexity or a comparison across different models (e.g., SDXL, DiT, and Flow Matching). Moreover, since the role of text is not effectively demonstrated, I believe that comparisons with text-guided image inpainting methods are highly unfair. Instead, comparisons with traditional diffusion-based inpainting methods such as RePaint, CoPaint, DDNM, StrDiffusion and IR-SDE would be much more reasonable.

Based on the above, I believe this submission requires a complete revision — including redoing the experiments and rewriting the manuscript — before resubmission to a venue

**Questions:**

- The details in Weaknesses

- The motivation of this paper needs to be restated, and the experimental section should better demonstrate the rationale and validity of the proposed motivation.

- The authors should clearly state whether the submission mainly addresses the problem of image inpainting or text-guided image inpainting, as the challenges faced by these two tasks are fundamentally different. Moreover, the paper fails to include a comparison with recent state-of-the-art work, such as CVPR 2025 “Towards Enhanced Image Inpainting: Mitigating Unwanted Object Insertion and Preserving Color Consistency.”

- Additionally, several essential ablation and detail experiments are missing and should be supplemented to strengthen the empirical analysis.

---

### Official Review · Reviewer_RJi9 · 2025-10-31

**Soundness:** 3
**Presentation:** 3
**Contribution:** 3
**Rating:** 6
**Confidence:** 3

**Summary:**

This paper introduces a method to align the training dynamics of both the VAE and the denoiser with the piece-wise latent manifold inherent in inpainting tasks. It proposes two key components: BlendRecon and BlendGen. BlendRecon ensures boundary continuity by enabling the VAE to correct mismatched mask gaps, while BlendGen simulates the blending operation during training to produce smoother image content. Additionally, the paper introduces Seam Visibility (SV) to measure stitched seams. Extensive experiments demonstrate that the approach significantly outperforms existing methods in both image quality and seam visibility, effectively reducing boundary artifacts.

**Strengths:**

The method is intuitively clear and well-motivated. The design is simple yet effective.
The proposed Seam Visibility (SV) metric offers a clear, quantitative way to evaluate seam quality, filling a gap in existing assessments.

**Weaknesses:**

The main concern is that key design choices and evaluation metrics lack clear alignment with their stated objectives. This raises questions about the individual contributions of the proposed components and the validity of the measurement. Please refer to the Questions section for more specific criticism and suggestions.

**Questions:**

In L238, Eq. 7 enables the decoder to learn seamless reconstructions of blended latents, thereby reducing boundary discontinuity. However, the fine-tuning objective in Eq. 7 applies a reconstruction loss over the entire image, which does not explicitly focus on the boundary regions where discontinuities occur.

In L346, seam visibility (SV) is defined as the average L2 distance of RGB values between pixels along the mask boundary and their neighboring pixels. However, the calculation of Eq. 8 is to process the pixels at the same position of the inpainted images and the original images.

In the ablation study, neither BlendRecon nor BlendGen alone yields a significant improvement in Seam Visibility (SV), yet their combination leads to a substantial reduction in SV. This raises two questions: (1) Why does the joint use produce a synergistic effect that neither component achieves individually? (2) Since BlendRecon is specifically designed to reduce boundary discontinuities, why does it not lead to a measurable improvement in SV when applied alone?

---

### Official Review · Reviewer_fdKh · 2025-11-01

**Soundness:** 2
**Presentation:** 2
**Contribution:** 1
**Rating:** 2
**Confidence:** 5

**Summary:**

This paper proposes a new LDM-based inpainting method. The core problem tackled in the paper is resolving the "stitch seam" problem in LDM-based inpainting. The main idea is to fine-tune the LDM components explicitly on blended inputs/latents. Two main components: (i) The VAE is fine-tuned on blended inputs. (ii) The denoiser is also fine-tuned on blended latents. However, the latter is done in a more carefully designed manner: The blended latents are constructed based on the prediction of $z_0$ where noise is then reapplied and true $z_0$ is subtracted to create the training target. Experiments demonstrate that the proposed method achieves state-of-the-art results on BrushBench and MISATO.

**Strengths:**

- The design of the proposed method is plausible and convincing.

- State-of-the-art results.

**Weaknesses:**

- The level of innovation is somewhat low. What the method basically does is fine-tuning the components explicitly on "seamed" data.

- A noticeable part is the training-target construction for the denoiser. I get the idea in this "detouring," since the denoiser must have a valid target and input for the seamed latents. However, a question still remains: Is it really necessary to go all the way to the estimate of $z_0$ (I understand that it is a single-step estimate)? It seems to me that a valid target can still be similarly (and more simply) constructed based on $z_t$, $z_0^M$, and predicted $z_{t-1}$.

[minor points]

- The propositions on page 4 are more like remarks rather than propositions. They are quite obvious, and I suggest changing the presentation here.

- The math notations in Algorithm 1 are somewhat poor. I suggest carefully reviewing them.

**Questions:**

Please see the above weaknesses.

---

### Official Review · Reviewer_xB6p · 2025-11-03

**Soundness:** 3
**Presentation:** 4
**Contribution:** 3
**Rating:** 6
**Confidence:** 4

**Summary:**

This paper introduces a new method for fine-tuning latent diffusion models, such as stable diffusion, for inpainting. The authors focus specifically on addressing a common issue with previous approaches, which is visible discontinuities between the inpainted regions of the image and the original image. The approach proposed by the authors introduces separate updates to fine tune the VAE decoder and the denoising network used in the latent diffusion model. The update to the VAE fine-tuning uses a reconstruction loss that specifically reconstructs from a blended encoding of a masked image and the full original image, comparing it to the full original image. This loss is designed to force the decoder to learn to produce continuous images even with discontinuous latent codes. The second contribution is a "blend aware denoiser" training procedure, which similarly fine-tunes the denoiser using blended inputs. In their experiments, the authors provide results on BrushBench and MISATO datasets, showing improvements in perceptual metrics such as FID and LPIPS, as well as a user study. The authors also introduce a new metric that measures the L2 loss compared to the original image specifically around the mask boundaries and show that their method improves this as well.

**Strengths:**

**Methodology**
- The proposed method is very simple, directly fine-tuning the existing model rather than using an auxiliary network, and simply adjusting the loss function to account for discontinuities in the inputs. It appears to be easy to implement and applicable other similar latent diffusion models.
- Despite its simplicity, it is novel to my knowledge.
- Adding a metric to measure reconstruction specifically around the mask boarder is interesting and could be useful for other works investigating techniques for inpainting.

**Results**
- The results appear to be strong. On standard metrics the approach outperforms other recent works in inpainting, sometime seemingly by a significant margin.
- The experiments are performed on recent, high-resolution benchmarks
- The authors include an ablation study that justifies the use of both contributions for their results
- The qualitative examples are compelling, clearly looking better than previous methods.

**Paper**
- The writing of the paper is mostly clear and well-motivated
- There are numerous qualitative examples shown in the main text

**Weaknesses:**

**Experiments**
- The version of stable diffusion (v1.5) used is now somewhat outdated. It makes sense as a benchmark against other methods implemented with that version, but it would be useful to see how this performs on more recent models with updated architectures.
- The experiments don't compare to conditional models for inpainting or control nets. Some of the cited comparisons do, so this is not a huge issue, but it would be a good update. Similarly, it would also be useful to compare to zero-shot approaches like [1,2, 3] etc.
- The corresponding prompts used for the results are not shown.

**Writing**
- It's not clear to me what the value of the theoretical insight section is, as the formalisms are not used elsewhere.
- The definition of seam visibility is unclear. The text states it's: "the average L2 distance of RGB values between pixels along the mask boundary and their neighboring pixels." but the equation suggests it's comparing the inpainted image to the ground truth.
- There is no discussion of the computation used for training the model
- There is no discussion of the potential negative social consequences of undetectable image editing models

[1] Zhang, Bingliang, et al. "Improving diffusion inverse problem solving with decoupled noise annealing." Proceedings of the Computer Vision and Pattern Recognition Conference. 2025.
[2] Rout, Litu, et al. "Solving linear inverse problems provably via posterior sampling with latent diffusion models." Advances in Neural Information Processing Systems 36 (2023): 49960-49990.
[3] Chung, Hyungjin, et al. "Diffusion posterior sampling for general noisy inverse problems." arXiv preprint arXiv:2209.14687 (2022).

**Questions:**

- Would this work with adaptor methods instead of fine-tuning the full model?
- Does the fine-tuned model still have equivalent performance in generating full images?
- How does the method perform with other types of masks. E.g. masks that are not as smooth and convex as the examples shown, such as dropping out random pixels?
- Would there be any advantage to also using the mask and/or masked image as a conditioning input to this method?
- Consider citing PixPerfect as concurrent work:
Yao, Yuan, et al. "PixPerfect: Seamless Latent Diffusion Local Editing with Discriminative Pixel-Space Refinement."

---

### Note · Authors · 2025-11-13

I have read and agree with the venue's withdrawal policy on behalf of myself and my co-authors.